# Crystal structure of the 4-hydroxybutyryl-CoA synthetase (ADP-forming) from nitrosopumilus maritimus
Jerome Johnson [1], Bradley B. Tolar[2,3], Bilge Tosun[1], Yasuo Yoshikuni [4], Christopher A. Francis [2], Soichi Wakatsuki [5,6] ✉ & Hasan DeMirci [1,7] ✉

The 3-hydroxypropionate/4-hydroxybutyrate (3HP/4HB) cycle from ammonia-oxidizing Thaumarchaeota is currently considered the most energy-efficient aerobic carbon fixation pathway. The *Nitrosopumilus maritimus* 4-hydroxybutyryl-CoA synthetase (ADP-forming; Nmar_0206) represents one of several enzymes from this cycle that exhibit increased efficiency over crenarchaeal counterparts. This enzyme reduces energy requirements on the cell, reflecting thaumarchaeal success in adapting to low-nutrient environments. Here we show the structure of Nmar_0206 from *Nitrosopumilus maritimus* SCM1, which reveals a highly conserved interdomain linker loop between the CoA-binding and ATP-grasp domains. Phylogenetic analysis suggests the widespread prevalence of this loop and highlights both its underrepresentation within the PDB and structural importance within the (ATP-forming) acyl-CoA synthetase (ACD) superfamily. This linker is shown to have a possible influence on conserved interface interactions between domains, thereby influencing homodimer stability. These results provide a structural basis for the energy efficiency of this key enzyme in the modified 3HP/4HB cycle of Thaumarchaeota.

Despite growing awareness of global climate change, carbon dioxide emissions are still increasing[1]. In order to address this critical issue, engineered autotrophic carbon-fixation cycles are being developed from a broad array of natural sources as a potential strategy to reduce atmospheric carbon[2,3]. Carbon fixation refers to the conversion of inorganic carbon to biologically useful organic compounds. This very ancient process has led to at least 7 different pathways in all three domains of life throughout evolution[2,4–7]. One of them, the 3HP/4HB cycle of Thaumarchaeota, is considered the most efficient aerobic carbon fixation cycle[8] and may be responsible for 1% of global carbon fixation[9], making it an intriguing candidate for these studies[10]. The 3HP/4HB cycle (Fig. 1) was first discovered in the archaeon *Metallosphaera sedula*[11–14]. In this cycle, carbon dioxide is fixed onto acetyl-CoA by an acetyl-CoA/propionyl-CoA carboxylase before being reduced into 3-hydroxypropionate. 3-hydroxypropionate is then joined with CoA by an (ADP-forming) 3-hydroxypropionyl-CoA Synthetase (Nmar_1309) before further carboxylation and succinyl-CoA formation (one of two useful precursor molecules). The succinyl-CoA is then reduced to 4HB, before another round of CoA ligation by Nmar_0206 and

continuation towards the formation of the final two products, namely two acetyl-CoA molecules (the second useful precursor molecule)[11,15]. The mechanisms for the increased efficiency of the thaumarchaeal 3HP/4HB cycle have been attributed to the unique energy requirements of the thaumarchaeal enzymes[8]. oxygen tolerance of the 4-hydroxybutyryl-CoA dehydratase, the bifunctional nature of crotonyl-CoA hydratase/3-hydroxypropionyl-CoA dehydratase and acetyl-CoA/propionyl-CoA carboxylase, and the phosphate conservation of the 4-hydroxybutyryl-CoA and 3-hydroxypropionyl-CoA synthetases (Nmar_0206 and Nmar_1309). Both synthetases, Nmar_0206 and Nmar_1309, are shown to be ADP-forming, as opposed to the AMP-forming alternatives present in the crenarchaeal 3HP/4HB cycle[8,11–14] (highlighted in Fig. 1). This phosphate conservation results in a reduced energetic burden on the cell, thus making them useful biological tools for engineering studies. Nmar_0206, the ADP-forming 4-hydroxybutyryl-CoA synthetase, catalyzes the conversion of 4HB and CoA to 4HB-CoA using the energy from a single dephosphorylation of ATP. This enzyme is a homodimer consisting of 5 shuffled subdomains common among members of the ACD superfamily: the ATP-grasp and lid domains,

[1]Department of Molecular Biology and Genetics, Koç University, Istanbul, Türkiye. [2]Department of Earth System Science, Stanford University, Stanford, CA, USA. [3]Department of Biology and Marine Biology, University of North Carolina Wilmington, Wilmington, NC, USA. [4]The US DOE Joint Genome Institute, Lawrence Berkeley National Laboratory, Berkeley, CA, USA. [5]Department of Structural Biology, Stanford University, Stanford, CA, USA. [6]Biosciences Division, SLAC National Accelerator Laboratory, Menlo Park, CA, USA. [7]Stanford PULSE Institute, SLAC, Menlo Park, CA, USA. ✉e-mail: soichi.wakatsuki@stanford.edu; hdemirci@ku.edu.tr

2 CoA-binding domains, and a CoA-ligase domain (Fig. 2a). The organization and linkages between these shuffled domains varies widely between ACD member proteins but the role this shuffling plays in protein stability requires further investigation. ACDs perform their function fundamentally different from their AMP-forming counterparts. AMP-forming Acyl-CoA synthetases, such as those found in *M. sedula*, initiate their reactions by passing through an adenylation reaction—covalently binding their substrate to AMP before the final thioesterification event[16,17]. Instead, the ADP-forming synthetases pass through a [substrate]-phosphohistidine intermediate formed with a single gamma phosphate of ATP, thereby conserving an intact ADP (Fig. 2b)[18]. In this study, we determined the structure of Nmar_0206 to 2.8 Å. We describe the structure of Nmar_0206, highlighting the unique linker loop, interfaces, and binding site that differentiate it from other ACDs. This may represent important scientific and engineering opportunities for improved carbon fixation.

## Results and discussion
### Overall crystal structure of Nmar_0206

The X-ray crystal structure of Nmar_0206 was determined to 2.8 Å resolution with two subunits within the Asymmetric Unit Cell. Each monomer consists of 624 out of 698 residues of the full length Nmar_0206, functional as a homodimer confirmed through the GalaxyGemini server (PDB: 8WZU) (Fig. 3). The resolved X-ray crystal structure of Nmar_0206 is a homodimer in the asymmetric unit, each monomer consists of 624 out of 698 residues of the full length Nmar_0206 (PDB: 8WZU) (Fig. 3). Like all ACDs, Nmar_0206 consists of shuffled domains conserved within the superfamily. ACD domain structure is usually described in comparison to the *E. coli* Succinyl-CoA which has a domain structure of [alpha(1,2)/beta(3,4,5)], whereas Nmar_0206 has the single protein chain [1-2-5-4-3][19] (Fig. 2). In other words, the subdomains 1, 2, 5, 4 and 3 of Nmar_0206 are connected in series from the N- to C-terminus. Subdomain 1 consists of a Rossman-like fold, while subdomains 2 and 5 are flavodoxin-like. The subdomains 1 and 2 of one monomer form a CoA-binding pocket with the subdomain 5, a CoA ligase domain, of the other monomer. Subdomain 2 contains an evolutionarily conserved H256 on a flexible swinging loop composed of small amino acids (residues S246-I266; Fig. 3), which homologous sites have been proposed to cross a distance of 31.5 Å to interact with bound ATP in the ATP-grasp domain[19,20]. It is stabilized by two power helices within subdomains 2 and 5 of monomers, which present their N termini near H256 (Fig. 3), generating a positively charged environment into which a reactive phosphate ion would be introduced but in which a structurally similar sulfate is likely found (Fig. 4)[21,22]. In contrast to most ACD structures within the PDB, domain 4 is attached to the C terminus of subdomain 5 by an 18 residue solvent exposed linker loop spanning 35 Å between W452-K469 (Fig. 5a). Additionally, an interface can be found between subdomains 4 and 5 of the two monomers (Fig. 6). Subdomains 3 and 4 together form an ATP-grasp unit, with subdomain 3 existing within subdomain 4 between residues 503 and 577. Although subdomain 3 is catalytically important in both the binding and transfer of phosphate from ATP to the phosphohistidine swinging loop[20], it was not present within the electron density map, suggesting free movement when unbound. Subdomain 4 comprises the ATP-binding site with ATP-interacting residues present (Supplementary Fig. 3).

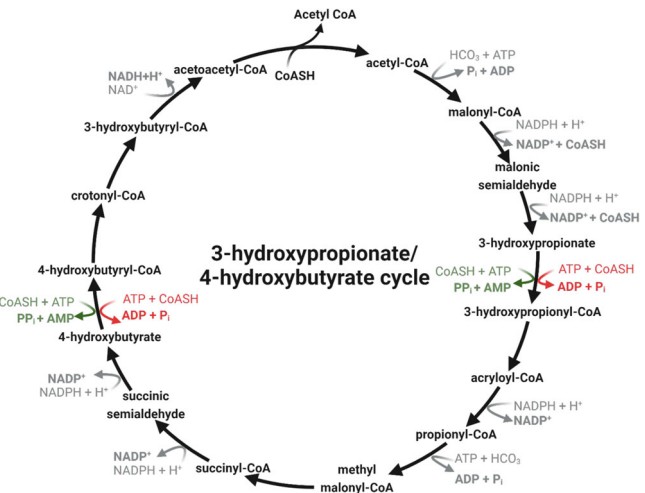

**Fig. 1 | 3HP/4HB cycle.** Green and red arrows indicate differences between the acyl-CoA synthetases of the 3HP/4HB cycles in Crenarchaeota and Thaumarchaeota, respectively. Both of these reactions are catalyzed by the respective 3-hydroxypropionyl-CoA synthetase and 4-hydroxybutyryl-CoA synthetase for each group, but Crenarchaeota enzymes are AMP-forming while Thaumarchaeota enzymes are ADP-forming.

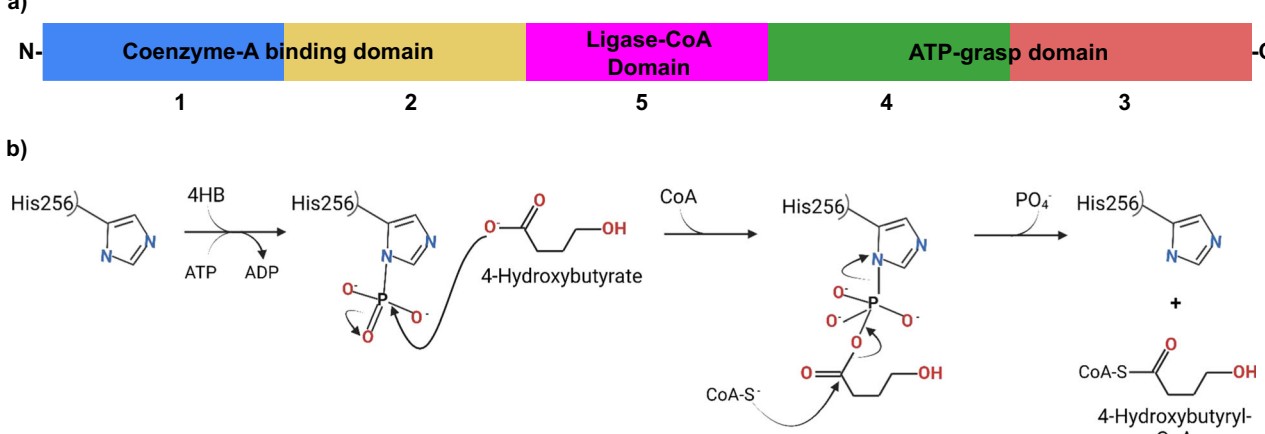

**Fig. 2 | Domain order and simplified reaction mechanism of Nmar_0206. a** The 5 subdomains of Nmar_0206 are differentiated by color. Subdomains 1 and 2 (blue and yellow), support CoA binding while 5 (pink), a ligase-CoA domain, supports the ligation of the 4HB group with CoA. The ATP-grasp domain contains subdomains 4 and 3 (green and red) with domain 3 functioning as a lid to close over a bound ATP. These domains are shown following *E. coli* Succinyl-CoA structural format [1-2-5-4-3], which can be compared with the *E. coli* Succinyl-CoA [alpha(1,2)/beta(3,4,5)] where alpha and beta indicate separate chains in that structure. The shuffling of subdomains is a definitive characteristic of the ACD superfamily. **b** The steps of this catalyzed reaction show intermediate phosphohistidine and formation of 4-hydroxybutyryl-phosphohistidine (near sulfinyl tail of 4HB at the interface between domains 1, 2 and 5) before final product formation is shown as part of a simplified reaction mechanism.

**Fig. 3 | Nmar_0206 (PDB: 8WZU) including important reactive elements, likely sulfate, and the linker between ATP-grasping and CoA-binding domains.** Green residues are proposed to interact with ATP (see pdb:4XYM) as part of the full reaction mechanism. The red colored swinging soop, which bridges the gap between ATP and the active site, transports a catalytically important histidine (yellow residue) to the tip of the two power helices. Blue power helices stabilize a likely sulfate found between the two helices (Fig. 4). Purple residues are proposed to interact with 4HB-CoA (see pdb:4XYM) as part of the reaction mechanism (see Supplementary Fig. 2).

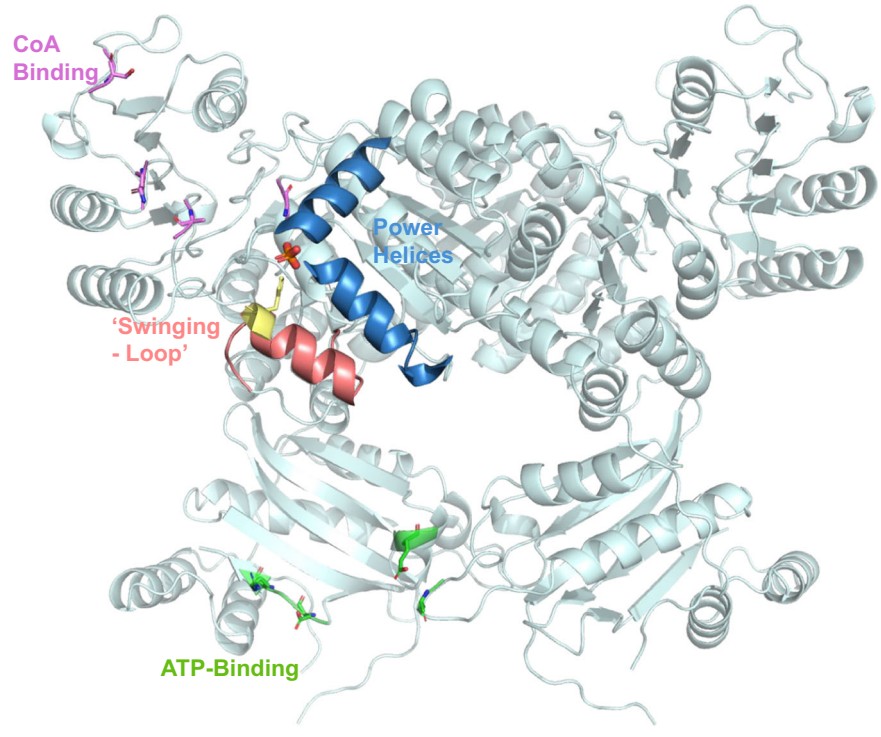

### Likely bound sulfate at active site of Nmar_0206

Although the Nmar_0206 structure lacks any of the standard interacting substrates, a positive electron density is found nearby H256 (side chain not found within the electron density). We posited that sulfate, structurally very similar to phosphate, from the crystallization buffer (including 200 mM lithium sulfate) is bound here and can be found interacting with G163, A164, G310 and G311 (Fig. 4). At the positive head of the two power helices, this sulfate could mimic the binding mode of phosphate, one of the products of the reaction. This configuration could represent a final leftover product bound to the structure after releasing 4HB-CoA and ADP.

### Linker loop and associated interfaces found between CoA-binding domain—ATP-Grasp domain

The presence of a similar linker loop between the two substrate binding macrodomains was first shown by Li et al.[21] in PDB 7CM9 from *Psychrobacter* sp. D2 (a homologous DMSP lyase). The Nmar_0206 structure similarly displays a 18 residue solvent exposed linker loop spanning 35 Å between W452-K469 (Fig. 5a, b) with an unclear function. A stark difference between the CoA-binding and ATP-grasp domain interfaces of the linker-containing Nmar_0206 and linker-less Acetyl-CoA synthetase structures may suggest that the linker provides an alternative to the stability that comes from hydrophobic domain interactions (Fig. 6b, d)[23,24]. Of the 138 ACD superfamily sequences used to generate the phylogenetic tree (Fig. 5c; Supplementary Data 1), 72.5% contain the aforementioned linker (five out of six representative groups). Within the PDB, structures exist for DMSP-lyase, succinyl-, citryl-, acetyl-, and now 4-hydroxybutyryl-CoA synthetases of which only two are heterotetrameric and do not contain a linker of some kind (see 7CM9 and 8WZU).

Within the greater ACD superfamily—enzymes that generally have roles in carbon fixation, acetate metabolism, and ATP generation—Nmar_0206 falls clearly into the Thaumarchaeota, as expected (Fig. 5c)[25]. The next closest relatives of Nmar_0206 are the methanogens (Methanobacteria, Domain Archaea), forming a distinct branch together with Crenarchaeota and Thaumarchaeota sequences. The other key enzyme of the 3HP/4HB cycle, Nmar_1309, shows some similarity to Nmar_0206 but is phylogenetically distinct; for this reason, Nmar_1309 and other related 3-hydroxypropionyl-CoA synthetase sequences can be seen as an outgroup

within our ACD tree (Fig. 5c). These phylogenetic relationships map to evolutionary separations between Archaea and Bacteria, and highlight the shift to the ACDs found within the 3HP/4HB cycle. This is supported in part by our results, as the closest relative to Thermoproteia ACDs (archaeal heterotetramers) are from Chloroflexota, a separate bacterial lineage (Fig. 5c). Such similarity between archaeal and bacterial ACDs would be unlikely to result from convergent evolution, potentially indicating horizontal gene transfer (HGT). As Thermoproteia ACDs are heterotetramers (including *Ca.* Korarchaeum, PDB: 4XYM; see also Fig. 6), these sequences might indicate an evolutionary link between homodimer Thaumarchaeota ACDs and the heterotetramer structures. This could suggest two separate HGT events: (1) between the heterotetramer Thermoproteia and the homodimer Chloroflexota, and (2) between the Chloroflexota and the rest of the homodimer Thaumarchaeota relatives. Even with this structural commonality, the amino acid sequence logos (Fig. 5b) display the large variability in amino acid composition of the linker loop.

Despite relative conservation in the interface between Nmar_0206 chains, the interface between the CoA-binding and ATP-grasp domains is much smaller in this homodimer structure when compared to the heterodimer acetyl-CoA synthetase (Fig. 6a–d). As the interface domain plays a role in dimerization, the smaller interface could be supported by these covalently fused chains. Assuming that this is the product of the linkage between the two domains, it is worth noting that *Ca.* Korachaeum typically live at high temperatures (78–92 °C)[26] whereas the mesophile *N. maritimus* grows at cooler oceanic temperatures (15–35 °C)[27]. The presence of this stabilizing linker loop in a mesophile and not a hyperthermophile gives further support to the suggested thermophilic ancestor of modern Thaumarchaeota[28]. However, it is intriguing that this feature also occurs throughout non-mesophilic Thaumarchaeota sequences (Supplementary Fig. 1).

### Residue shifts within homologous active sites

To accommodate a 4HB structure within the active site, some residues have shifted or been replaced. Using residues shown to be interacting with the *Ca.* Korachaeum acetyl-CoA synthetase as reference (PDB: 4YAK), comparisons were made between the Nmar_0206 structure (PDB: 8WZU) and others within the PDB with bound substrates (PDB: 6HXH and 5CAE) (Fig. 7b). Associated residues for L131, F146, A164, G355' and S385' can

**Fig. 4 | Power helices and a likely bound sulfate.**
Structures can be seen in the 2Fo-Fc electron density map with a contouring level of 1σ and carved to 2 Å. Power helices could stabilize a decoupled phosphate after dissociation from H256 within the reaction mechanism. H256 is not within the electron density map of this structure. Different shades of blue indicate different chains; yellow residues would interact with the sulfate, if bound.

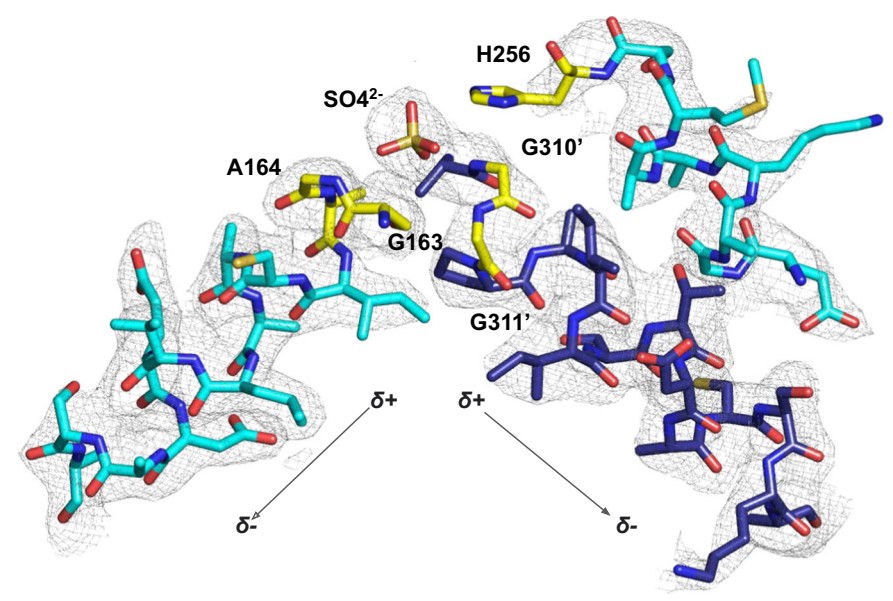

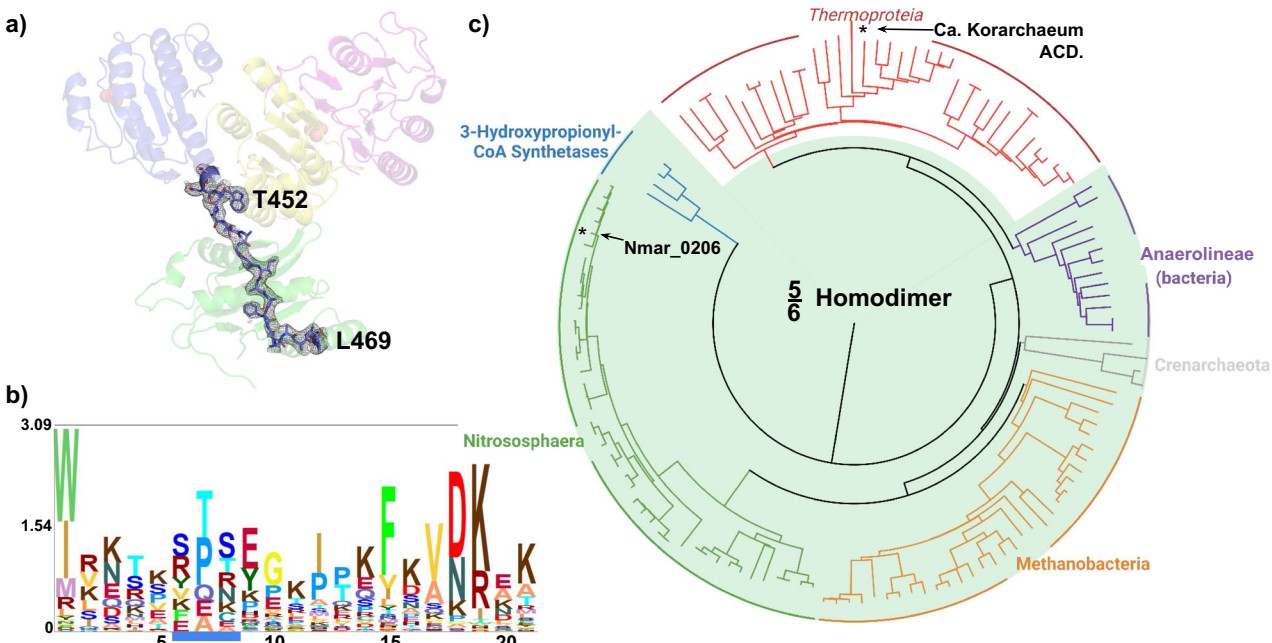

**Fig. 5 | Interdomain linker and phylogenetic tree. a** A 18-residue linker loop connecting the CoA binding (blue, yellow and pink representing subdomains 1, 2 and 5) and ATP-grasp domains (green representing subdomain 4) of Nmar_0206 can be seen in the 2Fo-Fc electron density map with a contouring level of 1σ and carved to 2 Å. **b** An amino acid sequence logos representation of the Nmar_0206 interdomain linker loop. Residues underlined in blue are not present in the Nmar_0206 sequence and have less than 10% representation within an MSA of 138 sequences of related structures (Supplementary Fig. 1). **c** Neighbor-joining phylogenetic tree containing 138 related ACD amino acid sequences from the NCBI protein database (same as MSA; Supplementary Data 1). Asterisks indicate two ACD structures present in the PDB including this Nmar_0206 structure and that of *Ca.* Korarchaeum ACD. Although additional ACD structures exist in the PDB, they are more distantly related and were thus excluded from this analysis (e.g., DMSP-lyase, succinyl- and citryl-CoA synthetases). The Nmar_0206 structure is one of only two homodimer ACDs in the PDB including 7CM9.

each be found within 5 Å of the acetyl group within 4YAK. With the exceptions of L131 and F146, these residues each shifted further away from that same location within the Nmar_0206 structure (Fig. 7a). The alpha carbons of S385', G355', and A164 each shifted by 2.5Å, 1.5Å and 1.1Å respectively.

A comparison among the related amino acid sequence logos (Fig. 7c) highlight residues originally identified as interacting with the acetyl-group from acetyl-CoA within PDB: 4YAK. The MSA (Supplementary Fig. 1) used to generate these logos show F146, A164 and G355 as being highly

conserved, suggesting important interactions with conserved elements (i.e., carboxyl end of substrate). L131 and S385 are much more variable, possibly indicating residues allowing for substrate promiscuity.

### Residue conservation within ACDs
Finally, we compared conservation in internal residues among ACDs between the subunits, particularly at the active site and ATP-binding sites (Fig. 8). Although all ACDs utilize CoA as a substrate, the ATP-binding site surprisingly appears only moderately conserved. Interface residues

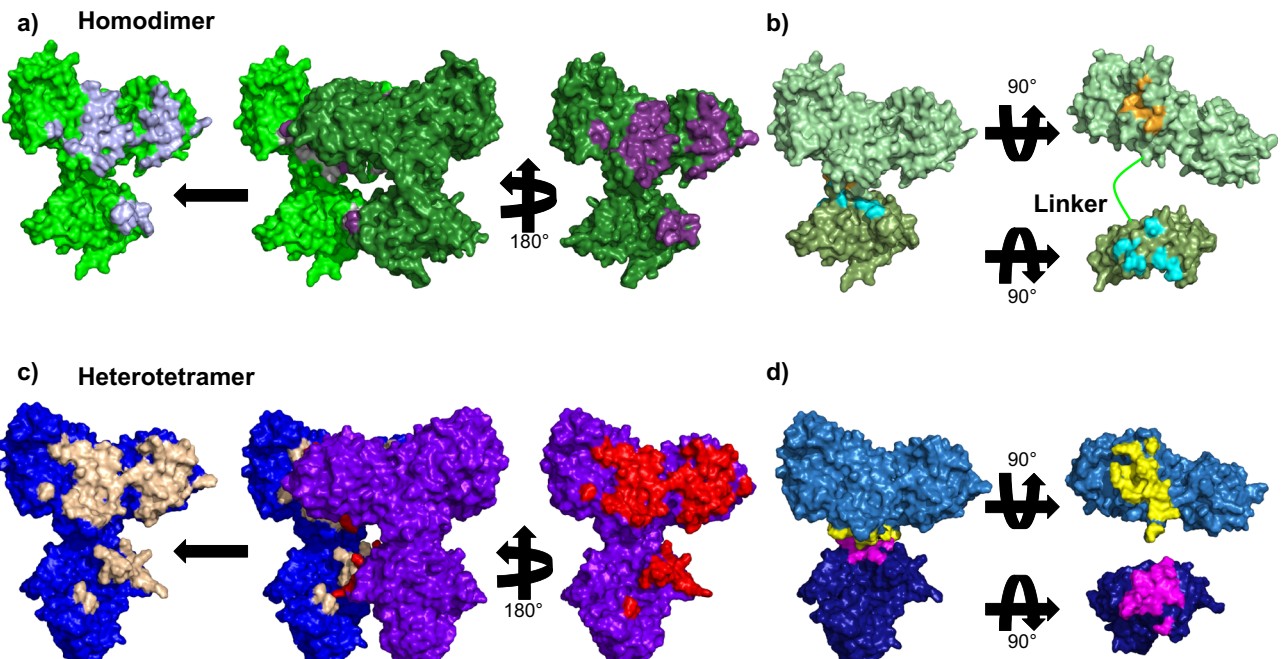

**Fig. 6 | Interface surfaces of Nmar_0206 and *Ca*. Korarchaeum Acetyl-CoA Synthetase (PDB: 4XYM).** Interfaces are displayed between the (**a**) homodimer Nmar_0206 monomers (gray and purple) and the **c** alpha domains and beta domains of *Ca*. Korarchaeum Acetyl-CoA Synthetase (tan and red). Interfaces are additionally highlighted between CoA-binding and ATP-grasp domains of (**b**) Nmar_0206 (orange and cyan) and (**d**) the *Ca*. Korarchaeum Acetyl-CoA Synthetase (yellow and pink). The homodimer Nmar_0206 (**b**) shows a reduced interface region when compared to the *Ca*. Korarchaeum Acetyl-CoA Synthetase (**d**).

(discussed in detail above) necessitate some conservation to maintain those structural connections. Although this is seen to some degree within the ConSurf results near the active site interactions (Fig. 8), there is still moderate variability in many residues along the interface.

### Likely reaction mechanism of Nmar_0206

The reaction mechanism for ACDs has been previously established[18–20,29]. The first of the reported reaction steps begins when the gamma phosphate of an ATP molecule is removed through a nucleophilic attack by a conserved H535 to form phosphohistidine[18,20] (Fig. 9). The phosphate should then be passed to the swinging loop's H256, which forms another phosphohistidine[19,20]. Phospho-H256 should then swing roughly 30 Å back towards the CoA-binding site between subdomains 1 and 2 where it binds to 4HB, forming a transient 4HB-phosphohistidine. The final step of the reaction mechanism involves the sulfur group from CoA performing a nucleophilic attack on the 4HB group, leading to the release of a free phosphate and 4HB-CoA as the final products[29]. Our structure, albeit missing the ATP-grasp domain lid (domain 3), is likely bound to sulfate within the active site, suggesting a structural conformation just following the reaction before the release of the free phosphate (Fig. 9; Supplementary Fig. 4).

### Materials & methods
#### Cloning

The *Nmar_0206* gene (see Supplementary Data 2) with an N-terminal histidine tag was designed and codon-optimized using Genscript BioTech trademark software before synthesis for Ni-NTA affinity purification. It was then cloned into the pET28a vector using *Nde*I and *Bam*HI endonuclease restriction sites, and transformed into *E. coli* strain BL21(Rosetta-2) purchased from Novagen. Transformed cells were selected for by growth in agar plates containing kanamycin (50 μg/ml) and chloramphenicol (50 μg/ml) at 37°C.

#### Protein expression and purification

Expression of Nmar_0206 was performed using the BL21(Rosetta-2) *E. coli* transformed cells. Cultures were grown overnight in LB media and then diluted 1:100 into 2 L cultures. The cell cultures were grown to an optical density of 0.8 at 600 nm and then expressed overnight at 18 °C following induction with 0.7 mM IPTG. Cell paste was then obtained following centrifugation at 3700 × *g*, resuspended in a lysis buffer (pH 7.0, 50 mM Tris, 300 mM NaCl, 5% v/v glycerol supplemented with 0.01% Triton X-100), and sonicated. Soluble protein was maintained at 4 °C, purified using Ni-NTA affinity resin (GE Healthcare) and concentrated to 10 mg/ml. The column was equilibrated with pH 7.0 HisA (containing 300 mM NaCl and 20 mM Tris). A column wash was performed using pH 7.0 HisA and eluted with pH 7.5 HisB containing 500 mM imidazole, 300 mM NaCl, 50 mM Tris. Following purification, thrombin and 5 mM beta-mercaptoethanol were added to Nmar_0206 to remove the N-terminal histidine tag and reduce any disulfide bonds that stabilize oligomers. Reverse Ni-NTA was then performed to remove the thrombin-cleaved hexa-histidine tag. Nmar_0206 purity was confirmed through SDS-PAGE. Following purification, 5 mL of 50% glycerol was added to 30 mL of the enzyme elutant for long-term storage.

#### Crystallization

Crystallization screens were performed using 72-well Terasaki microbatch plates. 0.83 μL of purified Nmar_0206 were pipetted manually into the bottom of the sitting drop well and mixed with an additional 0.83 μL of ~3500 commercially available sparse-matrix crystallization screening conditions[30] followed by a 16.6 μL application of Paraffin oil to protect samples from oxidation. Plates were stored in styrofoam containers at room temperature and crystals of Nmar_0206 were obtained 1-2 weeks after initial crystallization screenings. Protein crystals were obtained in a solution of 2.5 μL of 100 mM Na-acetate/acetic acid pH 4.5, 200 mM lithium sulfate, 50% (v/v) PEG 400, 2.5 μL of the enzyme solution, and 0.5 μL of 100% PEG 400 covered with ~50 μL of Paraffin oil.

#### Data collection and processing

Protein crystals of Nmar_0206 in 20% v/v glycerol were prepared for crystallography by flash freezing in liquid nitrogen. The data was collected at Stanford Synchrotron Radiation Lightsource Beam-Line 12-2 at SLAC, Menlo Park, California, USA. The detector distance for the enzyme

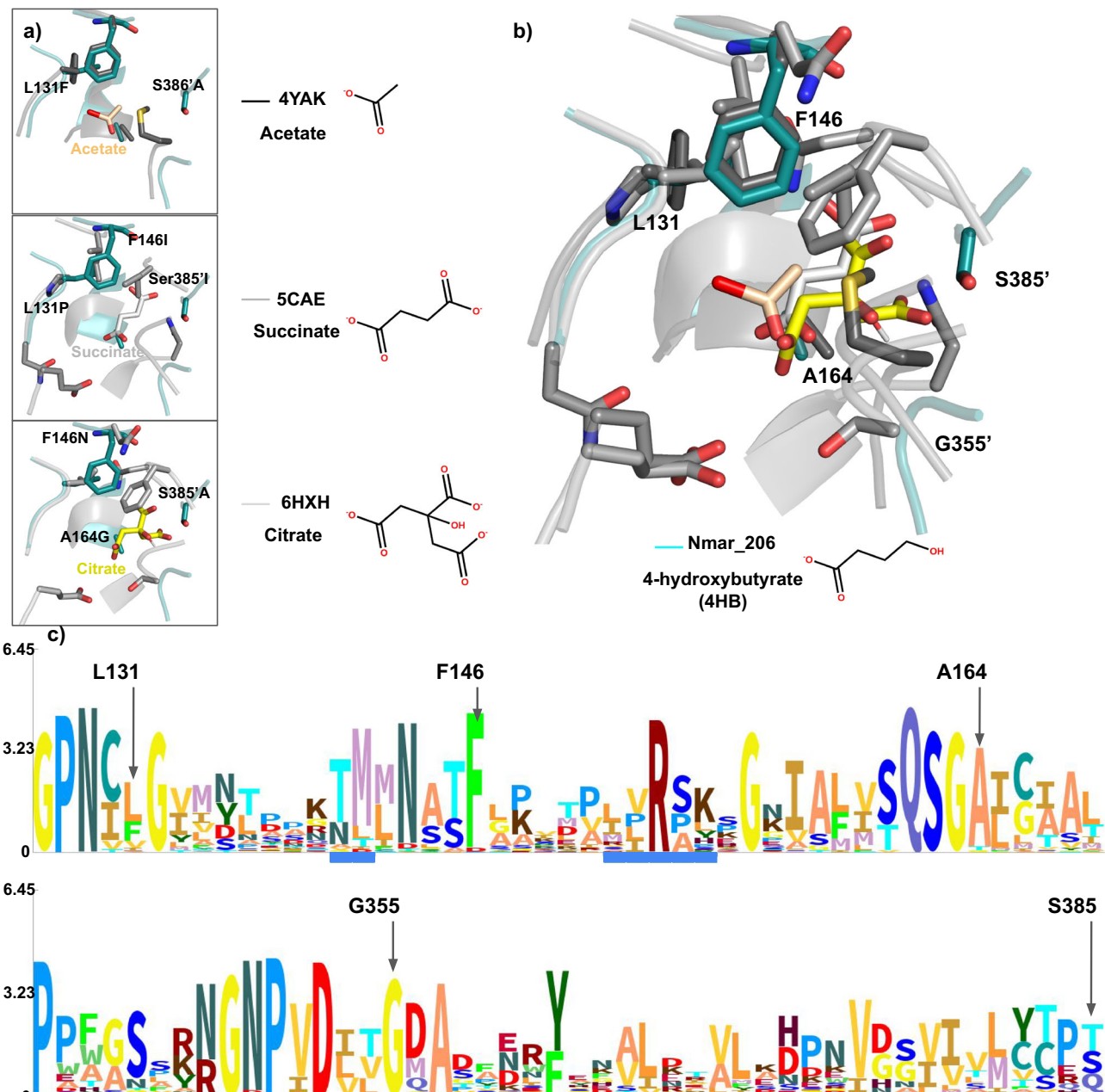

**Fig. 7 | Comparisons of Nmar_0206 with homologous active sites. a** Acetyl-CoA synthetase from *Ca*. Korarchaeum (PDB: 4YAK)[20], succinyl-CoA synthetase from *Sus domesticus* (PDB: 5CAE)[45], and citrate lyase from *Homo sapiens* (PDB: 6HXH)[46] show structural changes surrounding bound substrates. Substrates for Nmar_0206 (4HB), 4YAK (acetate), 5CAE (succinate), and 6HXH (citrate) are shown next to associated structures. Alignments were performed using the power helices tips and nearby projecting loops for reference, namely residues G310-P312, V354-D356, P128-C130, G163-I165, and G187-K189. The Pymol align algorithm was used with 0

outlier rejection cycles and provided RMSD values of 1.12 Å (4YAK), 1.61 Å (6HXH) and 0.77 Å (5CAE) based on their Ca residue positions. Residues within 5 Å of the carboxyl end of each substrate were displayed in stick mode. **b** Superposition of all structures highlight changes within the active site which may vary binding affinities to substrates. **c** Amino acid sequence logos for regions around the active site with labeled residues found interacting with acetate from acetyl-CoA synthetase (4YAK). For localization within the larger sequence, see Supplementary Fig. 1.

structure was set at 400 mm with an exposure time of 0.2 s per frame, with the X-ray energy set to 12.65 keV. The diffraction data were collected to 2.7 Å resolution at 100 K. The crystal belongs to the space group $P2_12_12_1$ with unit cell dimensions a = 70.53 Å b = 75.92 Å c =357.28 Å α = 90 β = 90 γ =90 [see Table 1]. The diffraction data were processed with the XDS package[31] for indexing and scaled by using XSCALE.

### Structure determination and refinement
A previously solved structure of ATP dependent dimethylsulfoniopropionate lyase (PDB ID: 7CM9) from *Psychrobacter* sp. D2[32] was used as a search

model for automated molecular replacement using *PHASER* within the *PHENIX* software suite[33]. This was followed by simulated-annealing, individual coordinate and TLS parameters refinement. COOT[34] was then used to confirm residue and water positions, and addition of residues missing in the electron density map. Further refinement was performed to 2.8 Å resolution, cut to a CC(1/2) of 0.43 with a completeness of 99.2%, within COOT while maintaining positions with strong difference densities and Ramachandran statistics for the structure were optimized to 94/5/00% (most favored/additionally allowed/disallowed) [see Table 1 for full refinement statistics]. The final R-work and R-free were 0.24 and 0.29

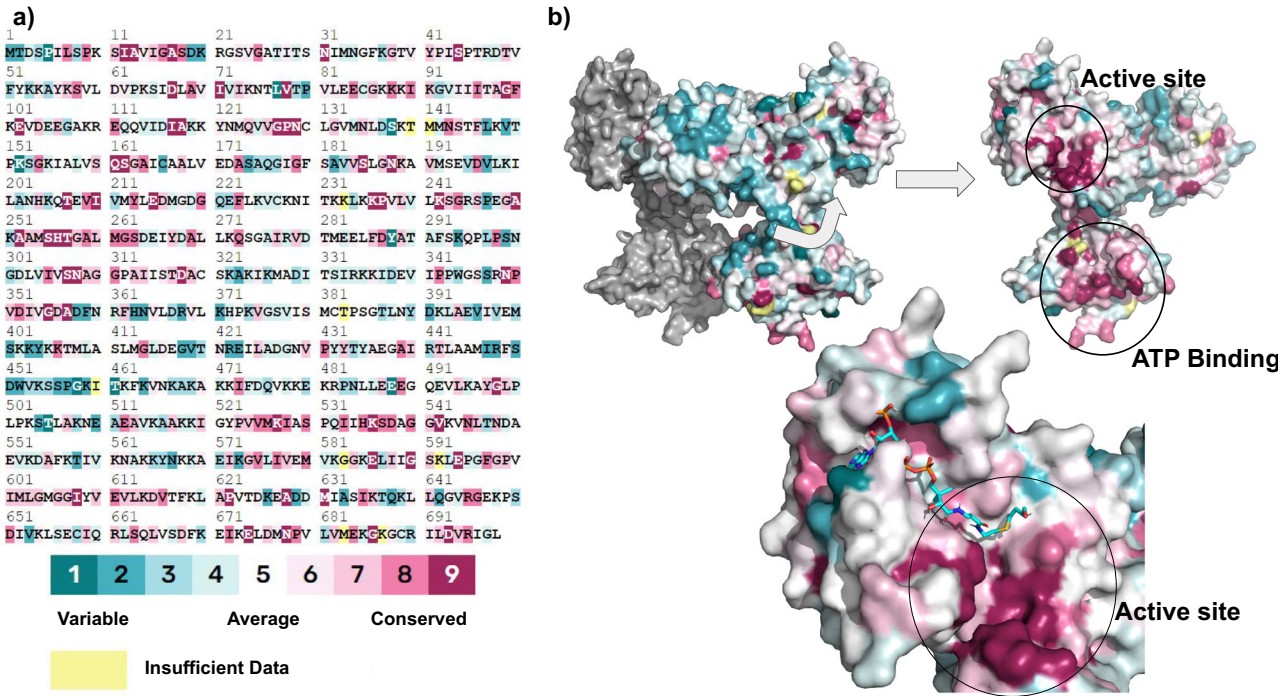

**Fig. 8 | ConSurf analysis for Nmar_0206.** Residue conservation is shown on the surface, coloring residues by conservation. 1, dark teal, indicates an extremely low conservation whereas 9, maroon, indicates a high sequence conservation. The information was obtained and is shown using the PyMOL output file from the ConSurf server[38]. **a** Residue are colored by conservation and (**b**) colors the surface conservation.

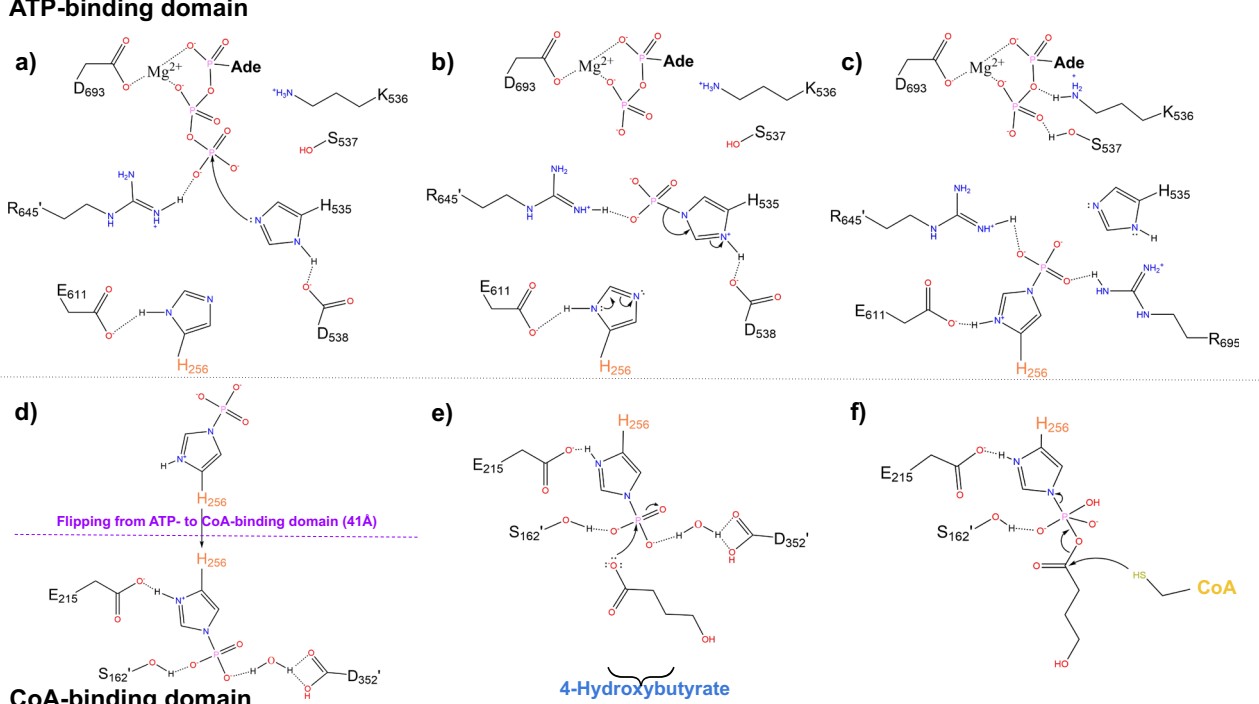

**Fig. 9 | Proposed reaction mechanism for 4HB-CoA formation from ATP, CoA and 4HB. a** The gamma phosphate of ATP is nucleophilically attacked by H535 before (**b**) being transferred onto H256, forming phosphohistidine. **c** Shows the result from these initial steps before (**d**) phospho-H256 crosses a distance of 30 Å into the active site. (**e**) It then interacts with 4HB to form (**f**) a transient 4HB–phospho-H256. The 4HB group is then attacked by the sulfinyl group of CoA leading to this dissociation of phosphate from H256 and the final formation of 4HB-CoA.

**Table 1 | Data collection and refinement statistics for Nmar_0206**

| PDB ID | 8WZU |
| --- | --- |
| Data collection | |
| X-ray source | *SSRL BL12-2* |
| Wavelength (Å) | *0.979* |
| Space group | P $2_1$ $2_1$ 2 |
| Cell dimensions | |
| *a, b, c (Å)* | 356.98 70.40 75.81 |
| α, β, γ (°) | 90.00 90.00 90.00 |
| Resolution (Å) | 24.87-2.69 (2.75-2.69) |
| CC (½) | 0.99 (0.27) |
| CC* | 0.99 (0.65) |
| I / σI | 7.49 (0.59) |
| Completeness (%) | 91.4 (75.3) |
| Redundancy | 13.16 (11.77) |
| Refinement | |
| Resolution (Å) | 24.9-2.8 (2.9-2.8) |
| No. reflections | 44,118 (3,542) |
| $R_{work}$ / $R_{free}$ | 0.24 / 0.28 (0.37 / 0. 41) |
| No. atoms | |
| Protein | 9397 |
| Ligand / Ion / Water | 93 |
| B-factors (Å²) | |
| Protein | 38.3 |
| Ligand / Ion / Water | 65.9 |
| Coordinate errors | 0.43 |
| R.m.s deviations | |
| Bond lengths (Å) | 0.013 |
| Bond angles (°) | 1.39 |
| Ramachandran plot | |
| Favored (%) | 93.5 |
| Allowed (%) | 4.6 |
| Disallowed (%) | 1.9 |

[1]The highest resolution shell is shown in parenthesis.

respectively. Structural figures were generated using PyMOL (version 2.5.2; Schrödinger) and Coot (version 0.9.8.1).

### Multiple sequence alignment preparation, phylogenetic tree generation and consurf

The NCBI Protein BLAST (blastp) was used to search for homologous sequences using Nmar_0206, Nmar_1309 and (ADP-forming) acetyl-CoA synthetase from *Candidatus* Korarchaeum as separate query templates (NCBI WP_012214589.1, WP_012215692.1 and WP_012308855.1 respectively). Out of the 300 sequences resulting from these three blastp searches, 138 unique amino acid sequences were selected based on species identifiers and used to construct a multiple sequence alignment (MSA) in MEGA (v11.0.13)[35] with the default settings for CLUSTALW program (v2.1)[36]. This MSA (Supplementary Fig. 1) was used to generate an amino acid-based phylogenetic tree of ACDs by the neighbor-joining method, using default settings within MEGA. The phylogenetic tree was then visualized with the iTOL: Interactive Tree of Life webserver[37]. The Consurf webserver was used to estimate evolutionary conservation at each amino acid position with default parameters and automatic homolog selection[38].

### Conclusion

The adaptation of a highly efficient 3HP/4HB cycle has helped *N. maritimus* and other ammonia-oxidizing Thaumarchaeota outcompete their bacterial counterparts in oligotrophic environments[39–41]. The Nmar_0206 structure represents just one component of the greater search for functional elements that differentiate the highly efficient 3HP/4HB cycle of Thaumarchaeota from less efficient variants represented in Crenarchaeota.

The ACD superfamily is used during carbon fixation, acetate metabolism, and ATP generation in many bacteria and archaea (see Supplementary Data 1)[25]. Synthetases specific to the 3HP/4HB cycle of Thaumarchaeota include Nmar_0206 and Nmar_1309. The homodimer seen within Nmar_0206 appears to be widespread within many microbial species, suggesting that this is not a unique development within Thaumarchaeota. Stability supported by the mentioned linker loop would therefore be relevant within many other structures within the ACD superfamily, suggesting two separate lineages of homodimers and heterotetramers; however, what evolutionary benefit such a fusion/separation would provide is still unclear.

To complete the overall goal of identifying structural components of increased energy efficiency, a similar analysis of an AMP-forming counterpart of 4HB-CoA synthetase from Crenarchaeota would need to be performed. Previous studies have described the AMP-forming substrate binding pocket as being amenable to mutation, due to its flexibility and the large size of its binding site[42]. Thus, substrate flexibility may partially explain the wide distribution of AMP-forming acyl-CoA synthetases (62 members in E.C. 6.2.1.) versus ACDs (9 members in E.C. 6.2.1.). Future investigations should perform comparative binding site analysis of AMP-forming crenarchaeal counterparts with Nmar_0206 to support this.

The enzymes involved in the 3HP/4HB cycle have been selected for energy-efficient attributes, such as the bifunctionality of crotonyl-CoA hydratase/3-hydroxypropionyl-CoA dehydratase[43] and the oxygen-tolerance of 4HB-CoA dehydratase[44]. The growing demand for carbon fixation strategies to remediate our rapidly changing atmosphere begs for further tools, such as those which can be produced from the characterization, and modification of highly efficient carbon cycling enzymes. The structural characteristics of Nmar_0206 and related enzymes could help inform scientists and engineers how to adapt these highly efficient enzymes for human needs.

### Data availability

The Nmar_0206 structure and associated files can be found in the PDB under 8WZU. The codon-optimized gene sequence can be found under Supplementary Data 2. All other supporting data are available from the corresponding author upon reasonable request.

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

## Acknowledgements

This project and the experiments are funded by TÜBİTAK-NSF 2501 bilateral research program (project number 221N355). H.D. acknowledges

support from NSF Science and Technology Center grant NSF-1231306 (Biology with X-ray Lasers, BioXFEL). This publication has been produced benefiting from the 2232 International Fellowship for Outstanding Researchers Program and the 1001 Scientific and Technological Research Projects Funding Program of the TÜBİTAK (Project Nos. 118C270 and 120Z520). However, the entire responsibility of the publication belongs to the authors of the publication. The financial support received from TÜBİTAK does not mean that the content of the publication is approved in a scientific sense by TÜBİTAK. The work conducted by the U.S. Department of Energy Joint Genome Institute (https://ror.org/04xm1d337), a DOE Office of Science User Facility, is supported by the Office of Science of the U.S. Department of Energy operated under Contract No. DE-AC02-05CH11231. SW and CAF acknowledge support from the U.S. Department of Energy (DOE) Office of Science, Biological and Environmental Research; Stanford Precourt Institute; and SLAC Laboratory Directed Research and Development. SSRL is supported by the U.S. Department of Energy (DOE), Office of Science, Office of Basic Energy Sciences (OBES) under Contract No. DE-AC02-76SF00515. The SSRL Structural Molecular Biology Program is supported by the DOE Office of Biological and Environmental Research and by the National Institutes of Health, National Institute of General Medical Sciences (NIGMS) (including P41GM103393). https://docs.google.com/document/d/15DKqmvS4hWirc7UkWO6nGVCpnu6zwXQnF-opz30YEp0/edit.

## Author contributions

H.D., Y.Y., C.A.F, S.W. conceived the study; H.D., J.J. designed the experiments; H.D., J.J., B.B.T., B.T. performed the experiments with input from S.W., J.J. and B.B.T. performed phylogenetic analyses. All the authors took part in the interpretation of results and preparation of the manuscript.

## Competing interests

The authors declare no competing interests.
