## [Peer Review File · Communications Biology]

Reviewers' comments:

Reviewer #1 (Remarks to the Author):

The manuscript by Johnson et. al. reports on the recombinant expression (from synthetic construct), purification, crystallization, and structure solution through X-ray crystallography of 4-Hydroxybutyryl-CoA Synthetase from *Nitrosopumilus maritimus*.

To this Reviewer the manuscript in its present form is largely incomplete and not suitable for publication.

The Results section is missing the crystallographic data collection and refinement statistics, which is normally provided as a table. For instance, the electron density shown in Figure 4 does not seem to appropriately fit the atomic model (by the way, the N atom of a proline residue has been accidentally dragged); this needs to be judged in light of the diffraction data and refinement quality. The authors need to provide the reflection intensity data on each resolution shell, data completeness, etc.

The Materials & Methods section is missing the bioinformatic tools and strategies reported in Figures 5 and 8, namely, phylogenetic analysis and Consurf analysis. Why was that particular set of sequences chosen for analysis? How were the multiple sequence alignment and phylogenetic tree calculated? What settings were selected for Consurf analysis and why?

Reviewer #2 (Remarks to the Author):

In this manuscript, Johnson et al report the crystal structure of 4-hydroxybutyryl-CoA synthetase from *Nitrosopumilus maritimus*. This enzyme is one of the specific enzymes of the HP/HB cycle that is responsible for its energetic efficiency, and the determination of its structure is important to gain molecular understanding of inorganic carbon fixation in ammonia oxidizing archaea (AOA). As the structure has relatively low resolution (2.7 Å) and was determined without substrates, the amount of new information that it can provide is limited. Nevertheless, analysis of the mechanism/active site is possible comparing the structure of Nmar_0206 with structures of homologous synthetases.

The manuscript also includes a phylogenetic analysis of homologous acyl-CoA synthetases, thus allowing to make some suggestions about the evolution of this enzyme family and the evolution of acyl-CoA synthetases in AOA.

Throughout the manuscript: 4-hydroxybutyryl-CoA synthetase is an acyl-CoA synthetase, and not an acetyl-CoA synthetase. Acetyl-CoA is a specific compound, while acyl-CoAs are a class of compounds. The family is therefore NDP-forming acyl-CoA synthetase family.

For the specificity of crenarchaeal synthetases: please better cite original literature, not Ref. 5 (Koenneke et al 2014), e.g. Berg et al., 2007; Alber et al., 2008, Ramos-Vera et al., 2011, or Hawkins et al, 2013, 2014. Please note that there is uncertainty concerning the gene encoding 4-hydroxybutyryl-CoA synthetase in *Metallosphaera* (though it is certainly an AMP-producing enzyme).

p. 3, legend: “The 5 subdomains of 4-Hydroxybutyryl-CoA”: “The 5 subdomains of 4-Hydroxybutyryl-CoA synthetase”

p. 5: “The Nmar_0206 gene was obtained from Genscript BioTech with a N-terminal histidine tag”: what does “obtained” mean? Was the gene synthesized? If yes, was codon usage optimized? What is the sequence of synthesized gene? If no, what were the primers, where *Nitrosopumilus* DNA was obtained?

p. 5: “Following purification, thrombin and beta-mercaptoethanol were added to Nmar_0206 to remove the N-terminal histidine tag and reduce any disulfide bonds that stabilize oligomers”: what was the concentration of mercaptoethanol? Was it necessary for the stability of the protein?

Was sulfate previously found in the synthetase structures?

Fig. 9 legend: 4-hydroxypropionyl-CoA: 4-hydroxybutyryl-CoA

p. 19: “This may suggest that the reason for the evolutionary preference of AMP-forming acetyl-CoA synthetases (62 members in E.C. 6.2.1.) over ACDs (9 members in E.C. 6.2.1.) is for substrate conversion despite their increased efficiency”: obviously, the reason for the choice of an AMP- or ADP-dependent enzyme is physiology (e.g., kinetics vs efficiency), and not a flexibility of the binding site. ADP-producing succinyl-CoA synthetase is a ubiquitously enzyme of the TCA cycle.

Reviewer #3 (Remarks to the Author):

The manuscript submitted by Johnson et al. reports the crystallographic structure of the enzyme 4-hydroxybutyryl-CoA synthetase (ADP-forming) from *Nitrosopumilus maritimus* (Nmar_0206) solved at 2.7 Å resolution. The manuscript focuses on analyzing the structure in comparison with homologs and concludes with a probable mechanism.

The structure reported is interesting and relevant to the field and some of the claims based on structural analysis are appropriate. However, based on the PDB validation report the quality of the data seems to be poor and the resolution limit should be set to a higher value. One major concern I have is about the sulfate ion the authors report and discuss about. The experimental data obtained is not enough to unequivocally identify this ligand as a sulfate, so all the results section and discussion about it should be removed, including the speculation about the reaction mechanism. The other interesting claim about the stabilizing connector loop is not carefully described in the text and not supported experimentally.

In general, the manuscript is not easy to read, there are major inconsistencies between the text

and the figures, especially figures not referenced in the text, and some words seem to be missing in some sentences. All this denotes a lack of consideration of the authors for the reader. Materials and Methods sections is incomplete, so experiments cannot be reproduced. A clearer, free of mistakes, and consistent version of the manuscript should be resubmitted.

Specific comments:

- 1- Rfree value seems too high, especially considering the authors used a test set of only 4% of the reflexions, the test set should be between 5 and 10%. Considering the reported $I/\sigma(I)$ value, it seems the structure should be solved at a lower resolution. The overall parameters from the PDB validation report seem to indicate that the refinement of the electron density map and the model could have been improved.
- 2- If the authors wish to keep the discussion about the sulfate ion they should provide some experimental evidence to identify this ion, such as sulfur SAD.
- 3- The connector loop should be described in more detail and stability should be determined experimentally if possible.
- 4- The text has mayor problems of consistency. Although it is not my job, I wrote down some corrections to be made: see points 5 to 9.
- 5- The name for each enzyme should be the same throughout the text.
- 6- Figure 1, 2, 3, 5A, 5B, 7A, 7B, and 9 are not referenced in the text.
- 7- Supplementary Figure 3 is mentioned but no supplementary data was provided.
- 8- At least the following details should be added to materials and methods section:
 - a. A table with the experimental crystallographic data statistics and refinement parameters must be appended to the manuscript.
 - b. 4000 rpm should be reported in g
 - c. Lysis and Ni-NTA equilibration buffers are not detailed
 - d. Beta-mercaptoethanol concentration is missing
 - e. Protein concentration in crystallization condition is missing
 - f. Cryo-protecting solution composition for flash freezing crystals is missing.
 - g. "ATP dependent dimethylsulfoniopropionate" should say "ATP dependent dimethylsulfoniopropionate lyase"
- 9- There is a mistake in the legend of Figure 6, some B) should be C).
- 10- 8IUP deposition at the PDB has a typo in the title.

Reviewer #1 (Remarks to the Author):

The manuscript by Johnson et. al. reports on the recombinant expression (from synthetic construct), purification, crystallization, and structure solution through X-ray crystallography of 4-Hydroxybutyryl-CoA Synthetase from *Nitrosopumilus maritimus*.

1. To this Reviewer the manuscript in its present form is largely incomplete and not suitable for publication. The Results section is missing the crystallographic data collection and refinement statistics, which is normally provided as a table. For instance, the electron density shown in Figure 4 does not seem to appropriately fit the atomic model (by the way, the N atom of a proline residue has been accidentally dragged); this needs to be judged in light of the diffraction data and refinement quality. The authors need to provide the reflection intensity data on each resolution shell, data completeness, etc.

*Thank you for the comment; we deeply apologize for this major oversight. Data collection and refinement statistics were added as follows as a table (see line 130). Additionally, the model has been updated with improved *r*-work and *r*-free values (up to 0.24 and 0.29 respectively). The mentioned proline residue 312 has been appropriately fixed in the figure as well (see line 203).*

Table 1. Data collection and refinement statistics for Nmar_0206.

PDB ID	8WZU
Data collection	
X-ray source	SSRL BL12-2
Wavelength (Å)	0.979
Space group	P 2 ₁ 2 ₁ 2
Cell dimensions	
a , b , c (Å)	356.98 70.40 75.81
α , β , γ (°)	90.00 90.00 90.00
Resolution (Å)	24.87-2.69 (2.75-2.69)
CC (1/2)	0.99 (0.27)
CC*	0.99 (0.65)
I / σ I	7.49 (0.59)
Completeness (%)	91.41 (75.30)
Redundancy	13.16 (11.77)
Refinement	
Resolution (Å)	24.87-2.80 (2.88-2.80)
No. reflections	44,119 (1,533)
R _{work} / R _{free}	0.24 / 0.29 (0.35 / 0.41)
No. atoms	
Protein	9397
Ligand / Ion / Water	144
B -factors (Å ²)	
Protein	54
Ligand / Ion / Water	52
Coordinate errors	0.43
R.m.s deviations	
Bond lengths (Å)	0.003
Bond angles (°)	0.63
Ramachandran plot	

Favored (%)	94
Allowed (%)	5
Disallowed (%)	0

¹The highest resolution shell is shown in parenthesis.

2. The Materials & Methods section is missing the bioinformatic tools and strategies reported in Figures 5 and 8, namely, phylogenetic analysis and Consurf analysis. Why was that particular set of sequences chosen for analysis? How were the multiple sequence alignment and phylogenetic tree calculated? What settings were selected for Consurf analysis and why?

The following was added to the Materials and Methods for MSA preparation and analysis due to constructive feedback (see lines 156-167):

Multiple Sequence Alignment Preparation, Phylogenetic Tree Generation and Consurf

The NCBI Protein BLAST (blastp) was used to search for homologous sequences using Nmar_0206, Nmar_1309 and (ADP-forming) acetyl-CoA synthetase from *Candidatus* Korarchaeum as separate query templates. Out of the 300 sequences resulting from these three blastp searches, 138 unique amino acid sequences were selected based on species identifiers and used to construct a multiple sequence alignment (MSA) in MEGA (v11.0.13)¹⁶ with the default settings for CLUSTAL W program (v2.1)¹⁷. This MSA (Supplementary Fig. 1) was used to generate an amino acid-based phylogenetic tree of ACDs by the neighbor-joining method, using default settings within MEGA. The phylogenetic tree was then visualized with the iTOL: Interactive Tree of Life webserver¹⁸. The Consurf webserver was used to estimate evolutionary conservation at each amino acid position with default parameters and automatic homolog selection.

Reviewer #2 (Remarks to the Author):

In this manuscript, Johnson et al report the crystal structure of 4-hydroxybutyryl-CoA synthetase from *Nitrosopumilus maritimus*. This enzyme is one of the specific enzymes of the HP/HB cycle that is responsible for its energetic efficiency, and the determination of its structure is important to gain molecular understanding of inorganic carbon fixation in ammonia oxidizing archaea (AOA). As the structure has relatively low resolution (2.7 Å) and was determined without substrates, the amount of new information that it can provide is limited. Nevertheless, analysis of the mechanism/active site is possible comparing the structure of Nmar_0206 with structures of homologous synthetases.

The manuscript also includes a phylogenetic analysis of homologous acyl-CoA synthetases, thus allowing to make some suggestions about the evolution of this enzyme family and the evolution of acyl-CoA synthetases in AOA.

1. Throughout the manuscript: 4-hydroxybutyryl-CoA synthetase is an acyl-CoA synthetase, and not an acetyl-CoA synthetase. Acetyl-CoA is a specific compound, while acyl-CoAs are a class of compounds. The family is therefore NDP-forming acyl-CoA synthetase family.

The document has been edited based on your feedback on this matter throughout the manuscript to accommodate the change. All references to 'acetyl-CoA synthetase superfamily' have been corrected to 'acyl-CoA synthetase superfamily.'

2. For the specificity of crenarchaeal synthetases: please better cite original literature, not Ref. 5 (Koenneke et al 2014), e.g. Berg et al., 2007; Alber et al., 2008, Ramos-Vera et al., 2011, or Hawkins et al, 2013, 2014. Please note that there is uncertainty concerning the gene encoding 4-hydroxybutyryl-CoA synthetase in *Metallosphaera* (though it is certainly an AMP-producing enzyme).

Thank you for these recommendations; we unintentionally overlooked this wealth of previous work beyond the Berg et al., 2007 reference, and have incorporated these in our discussion around crenarchaeal synthetases (see Lines 41, 48 and 56)

3. p. 3, legend: "The 5 subdomains of 4-Hydroxybutyryl-CoA": "The 5 subdomains of 4-Hydroxybutyryl-CoA synthetase"

Thank you for the suggestion, the figure legend has been updated to simply Nmar_0206 (see line 65).

4. p. 5: "The Nmar_0206 gene was obtained from Genscript BioTech with a N-terminal histidine tag": what does "obtained" mean? Was the gene synthesized? If yes, was

codon usage optimized? What is the sequence of synthesized gene? If no, what were the primers, where *Nitrosopumilus* DNA was obtained?

Thank you for the call for clarification. The gene was codon optimized using Genescript trademark software and can be found with other supplementary information (see lines 612-648). The manuscript has been updated as follows to explain in detail (see Lines 96-97):

A Nmar_0206 gene with an N-terminal histidine tag was designed and codon optimized using Genescript BioTech trademark software before synthesis for Ni-NTA affinity purification.

5. p. 5: "Following purification, thrombin and beta-mercaptoethanol were added to Nmar_0206 to remove the N-terminal histidine tag and reduce any disulfide bonds that stabilize oligomers": what was the concentration of beta-mercaptoethanol? Was it necessary for the stability of the protein?

To include the concentration, the manuscript has been updated at Line 112-114 as follows:

Following purification, thrombin and 5 mM beta-mercaptoethanol were added to Nmar_0206 to remove the N-terminal histidine tag and reduce any disulfide bonds that might lead to non-specific oligomerization.

6. Was sulfate previously found in the synthetase structures?

Previous articles have speculated on the presence of sulfate replacing phosphate at the tip of the power helices following crystallization in sulfate-containing conditions (see <https://pubs.acs.org/doi/10.1021/bi991696f>). As the crystallization condition contained 200 mM lithium sulfate (see line 126), we reasoned that the bound ion, very similar in structure and charge to phosphate, is likely sulfate

7. Fig. 9 legend: 4-hydroxypropionyl-CoA: 4-hydroxybutyryl-CoA

Thank you for the suggestion, the figure legend has been corrected as mentioned (see line 334).

8. p. 19: "This may suggest that the reason for the evolutionary preference of AMP-forming acetyl-CoA synthetases (62 members in E.C. 6.2.1.) over ACDs (9 members in E.C. 6.2.1.) is for substrate conversion despite their increased efficiency": obviously, the reason for the choice of an AMP- or ADP-dependent enzyme is physiology (e.g., kinetics

vs efficiency), and not a flexibility of the binding site. ADP-producing succinyl-CoA synthetase is a ubiquitously enzyme of the TCA cycle.

Thank you for the comment, it is the authors' opinion that although kinetics and efficiency are a central part to this discussion, they do not directly influence the wide variety of substrates that AMP-forming enzymes accommodate (62) when compared to ADP-forming enzymes (9). The passage has been updated as follows to further clarify that view (Lines 374-376):

Thus, substrate flexibility may partially explain the wide distribution of AMP-forming acyl-CoA synthetases (62 members in E.C. 6.2.1.) versus ACDs (9 members in E.C. 6.2.1.).

Reviewer #3 (Remarks to the Author):

The manuscript submitted by Johnson et al. reports the crystallographic structure of the enzyme 4-hydroxybutyryl-CoA synthetase (ADP-forming) from *Nitrosopumilus maritimus* (Nmar_0206) solved at 2.7 Å resolution. The manuscript focuses on analyzing the structure in comparison with homologs and concludes with a probable mechanism.

The structure reported is interesting and relevant to the field and some of the claims based on structural analysis are appropriate.

1. However, based on the PDB validation report the quality of the data seems to be poor and the resolution limit should be set to a higher value.

Thank you for the comment, the PDB has been resubmitted (PDB 8WZU) with a resolution limit of 2.8 improving the r-work and r-free to 2.4 and 2.9 respectively. The refinement statistics can be found in following table on line 130:

Table 1. Data collection and refinement statistics for Nmar_0206.

PDB ID	8WZU
Data collection	
X-ray source	SSRL BL12-2
Wavelength (Å)	0.979
Space group	P 2 ₁ 2 ₁ 2
Cell dimensions	
a , b , c (Å)	356.98 70.40 75.81
α , β , γ (°)	90.00 90.00 90.00
Resolution (Å)	24.87-2.69 (2.75-2.69)
CC (1/2)	0.99 (0.27)
CC*	0.99 (0.65)
I / σI	7.49 (0.59)
Completeness (%)	91.41 (75.30)
Redundancy	13.16 (11.77)
Refinement	
Resolution (Å)	24.87-2.80 (2.88-2.80)
No. reflections	44,119 (1,533)

R_{work} / R_{free}	0.24 / 0.29 (0.35 / 0.41)
No. atoms	
Protein	9397
Ligand / Ion / Water	144
B -factors (\AA^2)	
Protein	54
Ligand / Ion / Water	52
Coordinate errors	0.43
R.m.s deviations	
Bond lengths (\AA)	0.003
Bond angles ($^\circ$)	0.63
Ramachandran plot	
Favored (%)	94
Allowed (%)	5
Disallowed (%)	0

¹The highest resolution shell is shown in parenthesis.

2. One major concern I have is about the sulfate ion the authors report and discuss about. The experimental data obtained is not enough to unequivocally identify this ligand as a sulfate, so all the results section and discussion about it should be removed, including the speculation about the reaction mechanism.

Thank you for the insight. While we appreciate the call for skepticism here, previous articles have speculated on the presence of sulfate replacing phosphate at the tip of the powerhelices following crystallization in sulfate containing condition (see <https://pubs.acs.org/doi/10.1021/bi991696f>). As the crystallization condition contained 200 mM lithium sulfate (see line 126), we reasoned that the bound ion, very similar in structure and charge to phosphate, is likely sulfate. We have additionally moderated our language on the subject, using the words likely (see lines 171, 175, 205, 210, and 347); if (see line 208); and posited (see line 213) to be more cautious.

3. The other interesting claim about the stabilizing connector loop is not carefully described in the text and not supported experimentally.

Thank you for this note. To support our claim, references have been added to papers describing effects on loops on enzymatic thermal stability (see line 240 and references 34 and 35) and the text has been added to provide further discussion in lines 275-284.

The interface between the CoA-binding and ATP-grasp domains is much smaller in the homodimer structure when compared to the heterodimer (Fig. 6B, D). As the interface domain plays a role in dimerization, the smaller interface could be supported by these covalently fused chains. Assuming that this is the product of the linkage between the two domains, it is worth

noting that *Ca. Korarchaeum* typically live at high temperatures (78-92°C)²³ whereas the mesophile *N. maritimus* grows at cold to moderate oceanic temperatures (15-35°C)²⁴. The presence of this stabilizing linker loop in a mesophile and not a hyperthermophile gives further support to the suggested thermophilic ancestor of modern Thaumarchaeota⁴². However, it is intriguing that this feature occurs throughout non-mesophilic Thaumarchaeota sequences (Supplementary Fig. 1).

4. In general, the manuscript is not easy to read, there are mayor inconsistencies between the text and the figures, especially figures not referenced in the text, and some words seem to be missing in some sentences.

To resolve this discrepancy, references to all Figures have been added throughout the manuscript as per the reviewers comment. The following table details those:

Figure	Line #s
1	41, 56
2	80, 88,184
3	181, 189, 192
4	174 (Figure 3 Legend), 194, 215
5	196, 237, 241, 249, 254, 257, 265, 598
6	197, 240, 260, 276
7	306, 308, 312
8	327, 331
9	344, 352
Table 1	140, 153
Supplementary Figure 1	162, 225, 284, 299, 314
Supplementary Figure 2	202
Supplementary Table 1	226, 242, 362, 599

5. All this denotes a lack of consideration of the authors for the reader. Materials and Methods sections is incomplete, so experiments cannot be reproduced. A clearer, free of mistakes, and consistent version of the manuscript should be resubmitted.

Thank you for the constructive advice. To resolve the editors concerns, many changes have been made throughout the manuscript including a rewrite of a large section of the introduction (lines 76-91), an addition to the Materials and Methods to describe our sequence analysis work (lines 156-167), an additional discussion of phylogenetics (lines 247-266), along with numerous edits and word changes to improve readability.

Specific comments:

1- Rfree value seems too high, especially considering the authors used a test set of only 4% of the reflexions, the test set should be between 5 and 10%. Considering the reported $I/\sigma(I)$ value, it seems the structure should be solved at a lower resolution. The overall parameters from the PDB validation report seem to indicate that the refinement of the electron density map and the model could have been improved.

We appreciate the concern and have increased the test set used to 6.19% and 2979 reflections (bypassing the soft maximum of 2000 within phenix). This intern improved r -work and r -free values to 0.24 and 0.29 respectively. Regarding $I/\sigma(I)$, the current consensus seems to be that $CC(1/2)$ is a more appropriate measure for data quality within a particular bin for which we believe 2.8 Å (at 50%) is sufficient (see <https://doi.org/10.1016/j.sbi.2015.07.003>).

2- If the authors wish to keep the discussion about the sulfate ion they should provide some experimental evidence to identify this ion, such as sulfur SAD.

Previous articles have speculated on the presence of sulfate replacing phosphate at the tip of the power helices following crystallization (see <https://pubs.acs.org/doi/10.1021/bi991696f>). As the crystallization condition contained 200 mM lithium sulfate (see line 125), we reasoned that the bound ion, very similar in structure and charge to phosphate, is likely sulfate.

3- The connector loop should be described in more detail and stability should be determined experimentally if possible.

To resolve the reviewers comment, the manuscript has been updated with a new segment on the evolutionary origin of this structure (see lines 247-266):

Within the greater ACD superfamily - enzymes that generally have roles in carbon fixation, acetate metabolism, and ATP generation - Nmar_0206 falls clearly into the Thaumarchaeota, as expected (Fig. 5C)⁹. The next closest relatives of Nmar_0206 are the methanogens

(Methanobacteria), forming a distinct branch together with Crenarchaeota and Thaumarchaeota sequences. The other key enzyme of the 3HP/4HB cycle, Nmar_1309, shows some similarity to Nmar_0206 but is phylogenetically distinct; for this reason, Nmar_1309 and other related 3-hydroxypropionyl-CoA synthetase sequences can be seen as an outgroup within our ACD tree (Fig. 5C). These phylogenetic relationships map to evolutionary separations between Archaea and Bacteria, and highlight the shift to the ACDs found within the 3HP/4HB cycle. This is supported in part by our results, as the closest relative to Thermoproteia ACDs (archaeal heterotetramers) are from Chloroflexota, a separate bacterial lineage (Fig. 5C). Such similarity between archaeal and bacterial ACDs would be unlikely to result from convergent evolution, potentially indicating horizontal gene transfer (HGT). As Thermoproteia ACDs are heterotetramers (including *Ca. Korarchaeum*, PDB: 4XYM; see also Fig. 6), these sequences might indicate an evolutionary link between homodimer Thaumarchaeota ACDs and the heterotetramer structures. This could suggest two separate HGT events: (1) between the heterotetramer Thermoproteia and the homodimer Chloroflexota, and (2) between the Chloroflexota and the rest of the homodimer Thaumarchaeota relatives. Even with this structural commonality, the amino acid sequence logos (Fig. 5B) display the large variability of the linker loop's amino acid makeup.

4- The text has mayor problems of consistency. Although it is not my job, I wrote down some corrections to be made: see points 5 to 9.

These have been addressed, see below for specific details.

5- The name for each enzyme should be the same throughout the text.

The names for each enzyme have been corrected for consistency. Here is a list of the final acronyms used throughout:

Protein/Chemical/Cycle	acronyms used throughout	Lines found
4-hydroxybutyryl-CoA synthetase (ADP-forming)	Nmar_0206	11, 12, 13, 18, 46, 54, 65, 66, 76, 88, 89, 95, 102, 112, 114, 120, 123, 130, 134, 158, 171, 178, 179, 180, 181, 183, 185, 210, 211, 222, 223, 224, 227, 229, 236, 238, 248, 249, 252, 268, 269, 271, 272, 288, 291, 304, 308, 320, 341, 357, 363, 364, 374, 384, 593, 596, 605, 610, 612
3-hydroxypropionyl-CoA Synthetase	Nmar_1309	44, 54, 158, 251, 252, 363
(ATP-forming) acyl-CoA synthetase superfamily	ACD	22, 71, 79, 81, 82, 90, 163, 181, 182, 194, 226, 227, 228, 230, 241, 247, 254, 255, 256, 258, 259, 260, 261, 325, 326, 327, 342, 361, 366, 375
4-hydroxybutyrate	4HB	14, 46, 67, 72, 77, 176, 217, 291, 302, 334, 337, 339, 347, 349, 350, 371, 381, 597
3-hydroxypropionate/4-hydroxybutyrate cycle	3HP/4HB cycle	9, 27, 28, 38, 41, 49, 56, 251, 255, 355, 358, 362, 379

6- Figure 1, 2, 3, 5A, 5B, 7A, 7B, and 9 are not referenced in the text.

Think you for the reminder, each figure has been referenced to more easily guide readers.

Figure	Line #s
1	41, 56

2	80, 88, 184
3	181, 189, 192
4	174 (Figure 3 Legend), 194, 215
5	196, 237, 241, 249, 254, 257, 265, 598
6	197, 240, 260, 276
7	306, 308, 312
8	327, 331
9	344, 352
Table 1	140, 153
Supplementary Figure 1	162, 225, 284, 299, 314
Supplementary Figure 2	202
Supplementary Table 1	226, 242, 362, 599

7- Supplementary Figure 3 is mentioned but no supplementary data was provided.

For the purpose of revision supplementary figures were added to the bottom of the manuscript (see pages 31-39). These will be included as separate uploads for the final submission.

8- At least the following details should be added to materials and methods section:

a. A table with the experimental crystallographic data statistics and refinement parameters must be appended to the manuscript.

Thank you for the comment, we deeply apologize for the lack of oversight on this one. It has been added as Table 1 above Data Collection and Refinement in the materials and methods on line 130

Table 1. Data collection and refinement statistics for Nmar 0206.

PDB ID	8WZU
Data collection	
X-ray source	SSRL BL12-2
Wavelength (Å)	0.979
Space group	P 2 ₁ 2 ₁ 2
Cell dimensions	
a , b , c (Å)	356.98 70.40 75.81
α , β , γ (°)	90.00 90.00 90.00
Resolution (Å)	24.87-2.69 (2.75-2.69)
CC (1/2)	0.99 (0.27)
CC*	0.99 (0.65)

$I / \sigma I$	7.49 (0.59)
Completeness (%)	91.41 (75.30)
Redundancy	13.16 (11.77)
Refinement	
Resolution (Å)	24.87-2.80 (2.88-2.80)
No. reflections	44,119 (1,533)
R_{work} / R_{free}	0.24 / 0.29 (0.35 / 0.41)
No. atoms	
Protein	9397
Ligand / Ion / Water	144
B -factors (Å ²)	
Protein	54
Ligand / Ion / Water	52
Coordinate errors	0.43
R.m.s deviations	
Bond lengths (Å)	0.003
Bond angles (°)	0.63
Ramachandran plot	
Favored (%)	94
Allowed (%)	5
Disallowed (%)	0

¹The highest resolution shell is shown in parenthesis.

b. 4000 rpm should be reported in g

Thank you for the bringing up this issue, the manuscript has been updated to 3700g at line 106

c. Lysis and Ni-NTA equilibration buffers are not detailed

Lysis buffer and Ni-NTA equilibration buffers have been appended to the document at lines 105-107 as follows:

Cell paste was then obtained following centrifugation at 3700 g, resuspended in a lysis buffer (pH 7.0, 50 mM Tris, 300 mM NaCl, 5% v/v Glycerol supplemented with 0.01% Triton X-100) and sonicated.

And lines 108-109:

The column was equilibrated with pH 7.0 HisA containing 300 mM NaCl and 20 mM Tris.

d. Beta-mercaptoethanol concentration is missing

The manuscript has been updated at Lines 111-113 as follows:

Following purification, thrombin and 5 mM beta-mercaptoethanol were added to Nmar_0206 to remove the N-terminal histidine tag and reduce any disulfide bonds that stabilize oligomers.

e. Protein concentration in crystallization condition is missing

Thank you letting us know, the manuscript has been updated at lines 107-108 as follows:

Soluble protein was maintained at 4°C, purified using Ni-NTA affinity resin (GE Healthcare) and concentrated to 10 mg/ml.

f. Cryo-protecting solution composition for flash freezing crystals is missing.

The manuscript has been updated at Line 134-135 as follows:

Protein crystals of Nmar_0206 in 20% v/v glycerol were prepared for crystallography by flash freezing in liquid nitrogen.

g. “ATP dependent dimethylsulfoniopropionate” should say “ATP dependent dimethylsulfoniopropionate lyase”

Thank you for the comment, the manuscript has been updated as mentioned at Line 145.

9- There is a mistake in the legend of Figure 6, some B) should be C).

The manuscript has been updated as mentioned at Lines 269 and 271

10- 8IUP deposition at the PDB has a typo in the title.

Thank you for the comment, the title of 8IUP and the new deposition 8WZU have been updated to ‘4-hydroxybutyryl-CoA Synthetase (ADP-forming) from Nitrosopumilus maritimus.’

Reviewers' comments:

Reviewer #1 (Remarks to the Author):

General comments:

This manuscript describes the 2.80Å-resolution structure of Nmar_0206, the *N. maritimus* 4HB-CoA synthetase (ADP-forming). This structure comprises a marginally improved refinement of structure PDB 8IUP from a previous manuscript submission. Unfortunately, the structure has low resolution and no bound functionally-relevant ligands (only an ion modeled as a sulfate). Thus, the structure is comparable in quality with an AlphaFold2 model, with which it would have been interesting to compare. For instance, to see alternative conformations of the "swinging loop", the missing half of the ATP-grasp domain, the possible conformation of His256, and even modeling 4HB-CoA and ATP molecules by comparison with available structures.

A sulfate anion appears located between two "power helices", comparable with a phosphate in available homologous experimental structures, attributable to the location of a phosphate group upon catalytic turnover. Since several structures of acyl-CoA synthetases are tetrameric, with heterodimeric functional subunits of the form $\alpha\beta$, and Nmar_0206 is a single polypeptide with subunits α and β joined by a peptide linker, the authors attribute a (thermal) stabilizing effect for this linker.

Specific comments:

1. Page 1. I wouldn't call a 2.80 Å structure a "high-resolution" model.
2. Page 4. Please correct: "the ADP-grasp" should be "the ATP-grasp".
3. Page 6. Please mention how ~3500 conditions were tested by the microbatch method, were any liquid-handling robots used? Or trays were set manually?
4. Page 7. The authors provide now a (marginally) improved model for the crystallographic structure of 4HB-CoA Synthetase (ADP-forming) from *Nitrosopumilus maritimus* (Nmar_0206), as compared with the previous version (PDB ID 8IUP).
According to Table 1, X-ray diffraction data were collected to 2.69 Å resolution, and was cut to 2.80 Å resolution for model refinement. I agree with this, since *at 2.69 Å is lower than 1.0. However, the statistic, as well as CC(1/2), and completeness, at 2.80 Å, need to be reported too, in order to support this choice.*
Note that entry 8IUP should be obsoleted in the Protein Data Bank and replaced with entry 8WZU.
5. Page 8. Also, note that the manuscript on page 8 still incorrectly says "Further refinement was performed to 2.7 Å" and "The final R-work and R-free were 0.25 and 0.30 respectively."
6. Page 8. (Bioinformatics section) I could not find any sequence database entries for Nmar_0206 (NCBI, UniProt), which I believe is A9A1Y1 (witnessed from the 8IUP entry data). Please provide.
7. Page 10. The sentence "The resolved X-ray crystal structure of Nmar_0206 is a homodimer in the asymmetric unit, ..." needs to be reworded or split. The asymmetric unit (ASU) composition not necessarily indicates the protein quaternary structure (ie., homodimeric). Whereas the ASU is readily discernible from the crystal packing, the protein quaternary structure needs to be determined experimentally or estimated computationally, eg. with the PISA server.

Reviewer #2 (Remarks to the Author):

The authors have adequately addressed my comments in the revised version of the manuscript.

Reviewer #3 (Remarks to the Author):

General impression:

- 1- Overall readability of the manuscript has improved.*
- 2- Materials and Methods are now acceptably complete.*
- 3- Crystallographic data statistics are in the acceptable ranges.*
- 4- From the PDB validation report it is evident the model could be improved, at least the authors could run PDB-REDO on it.*
- 5- Although I find the claims about the sulfate ion highly speculative, language moderation in the text makes it acceptable.*
- 6- The description of the stabilizing linker has been improved.*

Specific comments:

- 1- "Structure determination and refinement" must be updated.*
- 2- There are mistakes in Supplementary Figure 2 legend.*
- 3- Since the authors did not provide the CC1/2 value in their original submission, they should be humbler in their answers tending to enlighten the reviewers about papers with thousands of citations.*

Reviewers' comments:

Reviewer #1 (Remarks to the Author):

General comments:

This manuscript describes the 2.80Å-resolution structure of Nmar_0206, the *N. maritimus* 4HB-CoA synthetase (ADP-forming). This structure comprises a marginally improved refinement of structure PDB 8IUP from a previous manuscript submission. Unfortunately, the structure has low resolution and no bound functionally-relevant ligands (only an ion modeled as a sulfate). Thus, the structure is comparable in quality with an AlphaFold2 model, with which it would have been interesting to compare. For instance, to see alternative conformations of the "swinging loop", the missing half of the ATP-grasp domain, the possible conformation of His256, and even modeling 4HB-CoA and ATP molecules by comparison with available structures.

We thank our reviewer for the suggestion. The new additions, supplementary figures 2 and 4, show two efforts made in this direction. Supplementary figure 4 shows the determined structures of intermediate states in relation to the reaction mechanism (see lines 626-635). Supplementary figure 2 shows 4-hydroxybutyryl-CoA placed within the binding site using the 4YAK structure as a guide, indicating possible interacting residues (see lines 608-611). While these comparisons have begun, we believed that the most pertinent comparison would be at the 4HB binding site (Fig.7 line 289) as this is the site at which substrates unique to different homologues bind.

Supplementary Figure 2 - 4HB-CoA placed into binding site. Using the 4YAK structure as a guide, 4HB-CoA was placed into its likely active site. Residues which could interact with the product are highlighted. The 4HB tail was manipulated looking for possible interactions for which Ser385 and Gly355 seemed possible candidates.

Supplementary Figure 4 - Structural representations of the ACD and Nmar_0206 reaction mechanism. Blue structures indicate those produced by Weiß et al. while green is this manuscript's Nmar_0206 structure. The 'Swinging-loop' is indicated in red. A) Shows the conformational state of the unbound apo-enzyme. B) Upon binding the ATP, the 'swinging loop' moves the conserved His256 nearby the ATP-grasping domain. C) The 'swinging loop' returns to the active site following the phosphorylation of His256 which may support the binding of further substrates. D) A structural intermediate showing the transient Acyl-phosphohistidine formation has yet to be determined. E) A structure with the bound Acyl-CoA product just prior to release. G) As phosphate is further within a pocket where the acetylation event takes place, the Nmar_0206 structure may represent a state just prior to the release of a final substrate.

A sulfate anion appears located between two "power helices", comparable with a phosphate in available homologous experimental structures, attributable to the location of a phosphate group upon catalytic turnover. Since several structures of acyl-CoA synthetases are tetrameric, with heterodimeric functional subunits of the form $\alpha\beta$, and Nmar_0206 is a single polypeptide with subunits α and β joined by a peptide linker, the authors attribute a (thermal) stabilizing effect for this linker.

Specific comments:

1. Page 1. I wouldn't call a 2.80 Å structure a "high-resolution" model.

We thank the reviewer for the clarification. The reference to a high-resolution structure has been removed on line 16 as follows:

Here we show the first ~~high-resolution~~ structure of Nmar_0206 from *Nitrosopumilus maritimus* SCM1, which reveals a highly conserved interdomain linker loop between the CoA-binding and ATP-grasp domains.

2. Page 4. Please correct: "the ADP-grasp" should be "the ATP-grasp".

We thank the reviewer for catching this. Line 77 has been corrected to reflect this as follows:

the ~~ADP-grasp~~ **ATP-grasp** and lid domains, 2 CoA-binding domains, and a CoA-ligase domain (Fig. 2a).

3. Page 6. Please mention how ~3500 conditions were tested by the microbatch method, were any liquid-handling robots used? Or trays were set manually?

We are grateful for the clarification, line 118 was update with the addition of 'manually' to resolve this obscurity:

0.83 µL of purified Nmar_0206 were pipetted **manually** into the bottom of the sitting drop and mixed with an additional 0.83 µL of ~3500 commercially available sparse-matrix crystallization screening conditions¹⁹,

4. Page 7. The authors provide now a (marginally) improved model for the crystallographic structure of 4HB-CoA Synthetase (ADP-forming) from *Nitrosopumilus maritimus* (Nmar_0206), as compared with the previous version (PDB ID 8IUP).

According to Table 1, X-ray diffraction data were collected to 2.69 Å resolution, and was cut to 2.80 Å resolution for model refinement. I agree with this, since at 2.69 Å is lower than 1.0. However, the statistic, as well as CC(1/2), and completeness, at 2.80 Å, need to be reported too, in order to support this choice.

Note that entry 8IUP should be obsoleted in the Protein Data Bank and replaced with entry 8WZU.

We are grateful for the advice. We are working with the PDB to update to the new PDB ID. Those values have been added to the Structure Determination and Refinement section on lines 148-151 as follows:

Further refinement was performed to 2.87 Å resolution, cut to a CC(1/2) of 0.43 ~~above 0.3~~ with a completeness of 99.2%, within COOT while maintaining positions with strong difference densities and Ramachandran statistics for the structure were optimized to 94/5/00% (most favored/ additionally allowed/disallowed) [see Table 1 for full refinement statistics]. The final R-work and R-free were 0.24 and 0.29 respectively.

5. Page 8. Also, note that the manuscript on page 8 still incorrectly says “Further refinement was performed to 2.7 Å” and “The final R-work and R-free were 0.25 and 0.30 respectively.”

We would like to thank the reviewer for the reminder, the document has been updated accordingly on lines 148 and 152 as follows:

Further refinement was performed to 2.87 Å resolution, cut to a CC(1/2) of 0.43 with a completeness of 99.2%, within COOT while maintaining positions with strong difference densities and Ramachandran statistics for the structure were optimized to 94/5/00% (most favored/ additionally allowed/disallowed) [see Table 1 for full refinement statistics]. The final R-work and R-free were ~~0.25 and 0.30~~ 0.24 and 0.29 respectively. Structural figures were generated using PyMOL (version 2.5.2; Schrödinger) and Coot (version 0.9.8.1).

6. Page 8. (Bioinformatics section) I could not find any sequence database entries for Nmar_0206 (NCBI, UniProt), which I believe is A9A1Y1 (witnessed from the 8IUP entry data). Please provide.

We are grateful for the addition. To make this more transparent, the following has been added to line 158:

The NCBI Protein BLAST (blastp) was used to search for homologous sequences using Nmar_0206, Nmar_1309 and (ADP-forming) acetyl-CoA synthetase from *Candidatus* Korarchaeum as separate query templates (NCBI WP_012214589.1, WP_012215692.1 and WP_012308855.1 respectively).

7. Page 10. The sentence “The resolved X-ray crystal structure of Nmar_0206 is a homodimer in the asymmetric unit, ...” needs to be reworded or split. The asymmetric unit (ASU) composition not necessarily indicates the protein quaternary structure (ie., homodimeric). Whereas the ASU is readily discernible from the crystal packing, the protein quaternary structure needs to be determined experimentally or estimated computationally, eg. with the PISA server.

We are grateful for the reminder. A correction has been made on lines 179-180 as follows:

The first X-ray crystal structure of Nmar_0206 was determined to 2.8 Å resolution with two subunits within the Asymmetric Unit Cell. Each monomer consists of 624 out of 698 residues of the full length Nmar_0206, functional as a homodimer confirmed through the GalaxyGemini server (PDB: 8WZU) (Fig. 3). ~~The resolved X-ray crystal structure of Nmar_0206 is a homodimer in the asymmetric unit, each monomer consists of 624 out of 698 residues of the full length Nmar_0206 (PDB: 8WZU) (Fig. 3).~~

We predicted the oligomeric state through the GalaxyGemini webserver. A link to those results can be found here:
https://galaxy.seoklab.org/cgi-bin/report_GEMINI.cgi?key=9340d3672cb036d7a86148b933dde27d

Reviewer #2 (Remarks to the Author):

The authors have adequately addressed my comments in the revised version of the manuscript.

Reviewer #3 (Remarks to the Author):

General impression:

1- Overall readability of the manuscript has improved.

- 2- *Materials and Methods are now acceptably complete.*
- 3- *Crystallographic data statistics are in the acceptable ranges.*
- 4- *From the PDB validation report it is evident the model could be improved, at least the authors could run PDB-REDO on it.*

In order to address this, PDB-REDO has been performed which resulted in moderate improvements to the overall structure. This has been submitted to the 8WZU pdb deposition. These included reduction of an unbiased R-free value from 0.30 to 0.28.

- 5- *Although I find the claims about the sulfate ion highly speculative, language moderation in the text makes it acceptable.*
- 6- *The description of the stabilizing linker has been improved.*

Specific comments:

- 1- *“Structure determination and refinement” must be updated.*

We are grateful for the reminder. Structural determination and refinement has been updated on lines 148-152 to reflect the improved structure as seen here:

Further refinement was performed to 2.87 Å resolution, cut to a CC(1/2) of 0.43 ~~above 0.3~~ with a completeness of 99.2%, within COOT while maintaining positions with strong difference densities and Ramachandran statistics for the structure were optimized to 94/5/00% (most favored/additionally allowed/disallowed) [see Table 1 for full refinement statistics]. The final R-work and R-free were ~~0.25 and 0.30~~ 0.24 and 0.29 respectively. Structural figures were generated using PyMOL (version 2.5.2; Schrödinger) and Coot (version 0.9.8.1).

- 2- *There are mistakes in Supplementary Figure 2 legend.*

We apologize for the misrepresentations. Supplementary Figure 3 title was updated to properly identify what it represents on Line 620. Residue numbers from the Weiße paper were added for comparison and ATP was corrected to ADPCP on line 623. Lines 620-624 are seen here:

Supplementary Figure 3 - ATP binding residues within Subdomain 3, the ATP-binding domain of

Nmar_0206 ~~Subdomain 4, the ATP-binding domain of Nmar_0206.~~ Although subdomain 4 is not present within our structure, residues Arg695 and Asp693 predicted by Weiße et. al (2016; **corresponding Arg226 and Asp224**) to interact with ATP are present. **ADPCP** ~~ATP~~ here (yellow) was copied from a superimposed pdb: 4XYM for rough estimation.

3- Since the authors did not provide the CC1/2 value in their original submission, they should be humbler in their answers tending to enlighten the reviewers about papers with thousands of citations.

We apologize for any offense given to the reviewer, particularly in our coming across without humility in our response discussing CC(1/2) after not initially reporting it. As we failed to provide the necessary statistics previously, we have corrected this oversight and now report CC(1/2) and I/ σ (I) at Lines 129 and 148. Additionally, we will endeavor to approach these conversations with a more delicate hand in the future and appreciate the reviewer's feedback on this subject.

Table 1. Data collection and refinement statistics for Nmar 0206.

PDB ID	8WZU
Data collection	
X-ray source	SSRL BL12-2
Wavelength (Å)	0.979
Space group	P 2 ₁ 2 ₁ 2
Cell dimensions	
a , b , c (Å)	356.98 70.40 75.81
α , β , γ (°)	90.00 90.00 90.00
Resolution (Å)	24.87-2.69 (2.75-2.69)
CC (1/2)	0.99 (0.27)
CC*	0.99 (0.65)
I / σ I	7.49 (0.59)
Completeness (%)	91.4 (75.3)
Redundancy	13.16 (11.77)
Refinement	
Resolution (Å)	24.9-2.8 (2.9-2.8)
No. reflections	44,118 (3,542)
R _{work} / R _{free}	0.24 / 0.28 (0.37 / 0.41)
No. atoms	
Protein	9397
Ligand / Ion / Water	93
B -factors (Å ²)	
Protein	38.3
Ligand / Ion / Water	65.9
Coordinate errors	0.43
R.m.s deviations	
Bond lengths (Å)	0.013
Bond angles (°)	1.39
Ramachandran plot	
Favored (%)	93.5
Allowed (%)	4.6
Disallowed (%)	1.9

¹The highest resolution shell is shown in parenthesis.

Further refinement was performed to 2.87 Å resolution, cut to a CC(1/2) of 0.43 above 0.3 with a completeness of 99.2%, within COOT while maintaining positions with strong difference

densities and Ramachandran statistics for the structure were optimized to 94/5/00% (most favored/ additionally allowed/ disallowed) [see Table 1 for full refinement statistics].

We therefore invite you to consider revising your paper one last time to address one remaining editorial concern. Portions of structures presented in some figures are perhaps too faint (e.g., it would be ideal to make the faintly-colored portions of the crystal structure in Fig. 3 a bit darker so that they can be seen more easily).

We'd like to thank the reviewers for this suggestion and have updated Fig. 3 to darken the main structure for visibility as follows: